# Salt-Related Knowledge, Attitudes, and Behaviors and Their Relationship with 24-Hour Urinary Sodium Excretion in Chinese Adults

**DOI:** 10.3390/nu14204404

**Published:** 2022-10-20

**Authors:** Fang Fan, Yinghua Li, Li Li, Xueqiong Nie, Puhong Zhang, Yuan Li, Rong Luo, Gang Zhang, Lanlan Wang, Feng J. He

**Affiliations:** 1School of Public Health, Anhui Medical University, Hefei 230032, China; 2Chinese Center for Health Education, Beijing 100011, China; 3The George Institute for Global Health, Beijing 100600, China; 4The George Institute for Global Health, Faculty of Medicine, University of New South Wales, Sydney, NSW 2042, Australia; 5Wolfson Institute of Population Health, Barts and The London School of Medicine and Dentistry, Queen Mary University of London, London EC1M 6BQ, UK

**Keywords:** knowledge, attitude, behavior, 24-h urinary sodium excretion, Chinese population

## Abstract

Salt intake in China is very high, which increases the risk of hypertension and cardiovascular disease. This study aimed to assess the levels of salt-related knowledge, attitudes, and behaviors (KABs) and the factors that influence them and to explore the relationship between the scores of salt-related KAB and 24-h urinary sodium excretion. In 2018, we collected data from 5453 individuals aged 18–75 years from six provinces in China. A face-to-face survey was carried out, focusing on the KAB related to salt reduction. All participants were asked to collect one 24-h urine sample. Of the 5453 participants, 5352 completed urine collection. The mean score for overall KAB was 31.27 (SD = 9.18), which was composed of three elements: knowledge 4.80 (SD = 5.14), attitude 9.33 (SD = 3.93), and behavior 17.14 (SD = 4.43). The average 24-h urinary sodium excretion was 187.70 (SD = 77.48) mmol, which was equivalent to a urinary sodium excretion of 4.32 (SD = 1.78) g/d. We found that salt-related knowledge, attitude, behavior, and overall KAB scores were all inversely associated with 24-h urinary sodium excretion. For every one-point increase in the KAB score, the 24-h urinary sodium excretion decreased by 0.851 mmol (95% CI: −1.095, −0.602). We also found that location (rural/urban), sex, age, and education are associated with salt-related KAB scores. These results suggest that large-scale health education is needed to reduce salt intake in the Chinese population. In particular, efforts should be focused on reaching those who live in rural areas with low educational levels and older people.

## 1. Introduction

Evidence has shown that a high salt intake raises blood pressure, which increases the risk of cardiovascular disease and stroke [1,2,3,4]. In 2017, more than 1.5 million deaths in China were attributable to a high-salt diet, making it the third most important risk factor for deaths and disability-adjusted life years [5]. Reducing the amount of salt in people’s diets is regarded as one of the most cost-effective public health measures for preventing noncommunicable diseases [6]. According to the latest large-scale 24-h urinary sodium study, the salt intake of the Chinese population was estimated to be 11 g/d [7]. To address the issue of excess salt consumption, the Chinese government has issued a series of salt-reduction-focused policies. In the ‘Healthy China 2030’ action plan, the government has set a goal of reducing salt intake by 20% by 2030 and recommends that all adults reduce salt intake to less than 5 g per day [8].

It is well known that the amount of salt intake mainly depends on individuals’ cultural backgrounds and eating habits. In many European countries, 75–80% of daily salt intake comes from processed foods [9]. In Japan, a large amount of salt (44%) comes from sauces and dressings [10,11]. In China, however, around 76% of the salt was added during cooking [12], and sauces account for about 9.8% of salt intake [13]. A previous study has shown that sauces sold in supermarkets in China contained more than four times the sodium compared to similar products in the United Kingdom [14].

The World Health Organization recommends that individuals’ knowledge and behaviors related to salt intake be assessed in order to develop interventions in line with the actual situation of the country [15]. Several studies have shown that consumer health education can effectively reduce salt intake [16,17]. In China, some small-scale assessments of salt-related knowledge, attitudes, and behaviors have been conducted in Beijing, Shandong, and Shanxi, most of which are carried out in the cities [17,18,19,20]. However, large-scale assessments of salt-related knowledge, attitudes, and behaviors, as well as 24-h urinary sodium data across the east, central, and west areas, are still lacking. Under the guidance and support of the National Health Commission of the People’s Republic of China, the Chinese Center for Disease Control and Prevention (China CDC), the Chinese Center for Health Education, the George Institute for Global Health (China), and the Queen Mary University of London had jointly launched Action on Salt China (ASC) [21]. ASC adopted a multi-sector approach to tackle the challenge of reducing salt intake in China [21]. ASC initiated four cluster randomized controlled trials (RCTs) in six provinces to develop and evaluate different salt reduction strategies, including the AppSalt-based salt reduction program for primary school children and their families (AIS) [22], the home cook salt reduction intervention study (HIS) [23], the community-based comprehensive salt reduction intervention study (CIS) [24], and the restaurant-based salt reduction intervention study (RIS) [25]. In this study, we collected baseline data from three RCTs, AIS, HIS, and CIS, in order to assess the level of knowledge, attitudes, and behaviors (KAB) related to salt intake in Chinese adults and to investigate the relationship between salt-related KAB and 24-h urinary sodium excretion. In addition, we assessed the differences in salt-related KAB with different socio-demographic characteristics (i.e., sex, age groups, and education levels) so as to provide evidence for the formulation of policies related to salt reduction.

## 2. Materials and Methods

### 2.1. Study Design and Participants

The overall design of Action on Salt China [21], as well as the specific design and data collection of the 3 RCTs (AIS [22], HIS [23], and CIS [24]), has been published before. Considering the geographical location, economic status, and eating habits, we selected study sites in six representative provinces (Hebei, Heilongjiang, Jiangxi, Hunan, Sichuan, and Qinghai) to conduct three randomized controlled trials of AIS, HIS, and CIS, respectively. Among the three RCTs, individuals aged between 18 and 75 with no relocation plans in the next two years were eligible, and those who were unable or refused to provide 24-h urine samples were excluded from the study. Eventually, AIS recruited 1184 adults and 592 children from 54 schools in three cities, HIS recruited 1576 participants from 60 communities/villages in six counties, and CIS recruited 2693 participants from 48 townships/streets in 12 counties.

### 2.2. Survey Instrument

We developed a new questionnaire due to no existing and validated one being available in China. The construct validity and the weight represented by the highest score for each question item were confirmed through three circles of expert consultations according to the importance/gaps of KAB for salt reduction in China. We then piloted it once among six family members of the researchers and twice among 15 children and 20 (grand-)parents in total in Shijiazhuang, with one RCT site of ASC to make sure that the questions were very well understood and the choices for each question had good differentiation.

The questionnaire contained 12 basic and representative questions, three of which pertained to knowledge, three to attitude, and six to behavior. Each question was worth 5 points, for a total of 60 points. Salt-related knowledge and attitude scores ranged from 0 to 15 points, respectively. For salt-related behavior questions, the scores ranged from 1 to 30. The higher the score, the higher the salt-related KAB.

Salt-related knowledge was assessed using the following three questions: (1) Do you know the recommended maximum level of salt intake per day (5 points if the answer is “less than 5 g” and 0 point for all other answers). (2) Have you heard of low-sodium salt (5 points if the answer is “yes” and 0 point for “no”). (3) Which item on the food label represents the salt content (5 points if the answer is “sodium” and 0 point for all other answers).

Salt-related attitudes were assessed using the following three questions: (1) Do you agree that high salt intake would cause hypertension (5 points if the answer is “agree” and 0 point for all other answers). (2) Do you agree that low salt intake would make people limb weakened (5 points if the answer is “disagree” and 0 point for all other answers). (3) Would you like to choose a lower-salt diet (5 points if the answer is “yes” and 0 point for all other answers).

Salt-related behaviors were assessed with six questions. (1) What is your usual taste for food (5 points if the answer is “less salty”, 3 points for ”moderate”, and 1 point for “more salty”). (2) Do you use low-sodium salt at home (5 points if the answer is “yes” and 0 point for all other answers). (3) The frequency of consuming pickled foods in the past month (5 points if the answer is “once a week or less”, 3 points for ”1–2 days a week”, 1 point for “3–5 days a week”, and 0 point for “almost every day”). (4) The frequency of consuming salty snacks in the past month (the same as question (3)). (5) The frequency of eating out or ordering delivery in the past month (5 points if the answer is “never”, 4 points for “once a week or less”, 3 points for ”1–2 days a week”, 1 point for “3–5 days a week”, and 0 point for “almost every day”). (6) Have you requested less-salted meals when eating out in the past month (5 points if the answer is “always”, 3 points for “sometimes”, 1 point for “occasionally”, and 0 point for “never”).

Blood pressure (BP) was measured by using a validated automatic machine. Three readings were taken in the right arm at 1–2 min intervals, and the average of the last two measurements was used to calculate mean BP. Hypertension was defined as mean systolic blood pressure (SBP) of ≥140 mm Hg or mean diastolic blood pressure (DBP) of ≥90 mm Hg or self-reported use of anti-hypertensive drugs in the previous two weeks. Participants with hypertension were divided into two groups, one for already diagnosed hypertension previously (old diagnosed) and the other for those observed at this screening (new observed).

Participants were also asked to collect urine samples for a 24-h period. The urine collection was excluded if the 24-h urine volume was less than 500 mL or the creatinine was less than 4.0 mmol for females or 6.0 mmol for males [20,26,27]. If the urine collection lasted less than 20 h or more than 28 h, it was also excluded [7]. Estimated urinary sodium excretion (g/24 h) was calculated from 24-h urinary sodium excretion using the formula: 1 mmol Na = 1 mEq Na = 23 mg Na, and urinary sodium excretion (g/24 h) = Na (mmol/24 h)·23/1000.

### 2.3. Data Analysis

Mean and SD were used to describe continuous variables and frequency, and percentages were used to describe the categorical variables. For comparison, age was combined into three groups, i.e., 18–44, 45–59, and ≥60. Education level was divided into three categories, low was defined as primary school education or less (0–6 years), medium was defined as junior high school education (7–9 years), and high was defined as senior high school or above (≥10 years). We used t/F tests to compare population groups across salt-related knowledge, attitude, behavior, and overall KAB scores. Considering that the participants in the three randomized controlled trials might have heterogeneity and potential clustering in the community, we established a mixed effect model to analyze the relevant factors affecting the scores of salt-related knowledge, attitude, behavior, and overall KAB by defining the random effects of randomized controlled trials and community level. Three models were established when examining the relationship between the scores of salt-related KAB and 24-h urinary sodium excretion. Model 1 was an unadjusted result. Model 2 adjusted for sex and age groups, and model 3 adjusted for location (rural/urban), sex, age groups, education levels, and hypertension (no/old diagnosed/new observed). All data were collated and analyzed using the statistical program IBM SPSS Statistics version 22, except for where R 4.2.0 was applied to the mixed effect models. A *p*-value of <0.05 was considered statistically significant.

## 3. Results

A total of 5453 adult participants were recruited in three RCTs. We excluded 101 incomplete urine samples, 10 for the 24-h urine volume < 500 mL, 52 for creatinine < 6.0 mmol for males, 33 for creatinine < 4.0 mmol for females, and six for both urine collection time and creatinine data missed. The remaining 5352 participants (98.1% of recruited participants; Figure 1) were included.

### 3.1. Demographic Characteristics of Participants

Of the 5352 participants, 3915 (73.2%) were from rural areas. The average age of the participants was 49.76 (SD = 12.84) years, 51.4% were females, and more than one-third (34.4%) had hypertension. The mean score for overall salt-related KAB was 31.27 (SD = 9.18), which was composed of three elements: knowledge 4.80 (SD = 5.14), attitude 9.33 (SD = 3.93), and behavior 17.14 (SD = 4.43). The average 24-h urinary sodium excretion was 187.70 (SD = 77.48) mmol/d, which was equivalent to a urinary sodium excretion of 4.32 (SD = 1.78) g/d (Table 1).

### 3.2. Knowledge, Attitudes, and Behaviors Related to Salt Intake

The awareness rate of the recommended maximum level of salt, ‘heard of’ low-sodium salt, and the salt nutrition label were 22.38%, 30.61%, and 43.07%, respectively. Of the participants, 76.21% agreed that high salt intake would cause hypertension, 28.27% believed that low salt intake would not make people weakened, and 82.01% were willing to choose a lower-salt diet. About one-third of the participants (29.65%) preferred salty tastes, while only 10.39% of the participants used low-sodium salt at home. A total of 39.29% and 8.11% of the participants ate pickled foods and salty snacks at least once a week, respectively. Nearly 80% of the participants reported that they had eaten out, while less than one-fifth of the participants (17.10%) had requested for less-salted meals when eating out (Table 2).

### 3.3. Knowledge, Attitude, and Behavior Scores Related to Salt

In general, salt-related KAB differed among those participants with different socio-demographic characteristics (Table 3). The salt-related KAB score was 35.59 ± 8.83 for the urban participants and 29.64 ± 8.81 for the rural participants (*p* < 0.05 rural vs. urban). Males had significantly lower salt-related attitude scores (8.79 ± 4.14 vs. 9.83 ± 3.66), behavior scores (16.54 ± 4.49 vs. 17.67 ± 4.32), and KAB scores (30.27 ± 9.35 vs. 32.13 ± 8.96) than females (all *p* < 0.001). In terms of different levels of education, the higher the level of education, the higher the score of salt-related knowledge, attitude, behavior, and KAB (all *p* < 0.05). For different age groups, the participants aged 18–44 years had the highest salt-related knowledge score, attitude score, and KAB score, followed by those aged 45–59 years and those over 60 years had the lowest. However, the salt-related behavior score was the opposite (all *p* < 0.05). For different hypertensive conditions, the normotensive individuals had the highest salt-related knowledge, attitude, and KAB scores; however, the hypertensive individuals who were diagnosed previously had a higher salt-related knowledge score but lower salt-related attitude and behavior scores than those newly observed (all *p* < 0.05).

### 3.4. Univariate and Multivariate Analysis of Salt-Related KAB Score

Age, sex, education, location (rural/urban), and hypertension (no/old diagnosed/new observed) were the influencing factors of the salt-related KAB scores, according to univariate analysis(Table 4 and Appendix A). Further multivariate analysis showed that location, sex, and education continued to be the determining factors of salt-related KAB scores even after adjusting for other factors. Keeping all other factors constant, participants who lived in rural areas had lower salt-related KAB scores than those who lived in urban areas by 1.165 (95% CI: −2.488, −0.150) points. Females had a higher salt-related KAB score than males by 2.734 (95% CI: 2.291, 3.186) points. Compared to the participants with primary school education or less, those with junior high school education had a higher salt-related KAB score by 3.802 (95% CI: 3.255, 4.342) points. Those with senior high school education or above had a salt-related KAB score that was 9.266 (95% CI: 8.628, 9.902) points higher (*p* < 0.001) (Table 4).

### 3.5. Association of Salt-Related Knowledge and Attitude Scores and the Behavior Score

In univariate and multivariate analysis, salt-related knowledge and attitude scores were both influencing factors over the behavior score, although the effects were modest (Table 5).

### 3.6. Association of Salt-Related KAB Score and 24-h Urinary Sodium Excretion

In the unadjusted model, the scores for each component of salt-related knowledge, attitude, and behavior were strongly correlated with 24-h urinary sodium excretion (all *p* < 0.001) (Table 6). In model 2, 24-h urinary sodium excretion was still negatively associated with the scores for salt-related KAB after controlling for age and sex, although the relationship weakened. In model 3, after further adjustment for location, education, and hypertension (no/old diagnosed/new observed), every 1-point increase in the score for salt-related behavior was linked to a mean reduction in the 24-h urinary sodium excretion by 2.200 mmol/d (95% CI: −2.673, −1.716). For every 1-point increase in the score for salt-related KAB, the 24-h urinary sodium excretion decreased by 0.851 mmol/d (95% CI: −1.095, −0.602) (Table 6).

## 4. Discussion

Our study, for the first time, has investigated the salt-related KAB and its relationship with salt intake, as measured by the most accurate method of 24-h urine collection from a large number of individuals in diverse settings within China. Our results indicated that salt-related knowledge, behavior, and overall KAB scores were all inversely associated with 24-h urinary sodium excretion. In other words, individuals who had a higher KAB score had a lower salt intake. This suggested that improving the level of salt-related KAB might be an effective way to reduce population-wide salt intake in China.

Generally speaking, Chinese people had a low level of salt-related knowledge. Consistent with previous studies [28,29,30,31], we found that location (rural or urban), sex, age, and education were associated with salt-related KAB scores among Chinese people. In rural areas, salt-related KAB scores were significantly lower than those in urban areas, which was consistent with the national conditions in China [32]. Education level demonstrated a significant influence on knowledge, attitude, and behavior related to salt reduction, which was in line with the findings by Chen [30] and Grimes [31]. Participants with a low education level had poor reading, and comprehension skills and their social status and work might make it more difficult for them to obtain health information resources. Contrary to the findings reported in 2018 [33], we also found that males had better knowledge of salt reduction than females. However, it was discovered that females have more favorable attitudes and behaviors regarding salt. In fact, females were more health conscious overall and were more likely to read and follow nutritional guidelines than males [34,35]. According to our results, younger people had better knowledge and attitude scores but worse behavior scores than older people. One explanation is that younger people are well-educated and have more access to health information, whereas older people are more likely to adopt healthy lifestyles, such as avoiding eating out or ordering deliveries [36,37]. The hypertensive individuals who were diagnosed previously had higher knowledge scores but lower attitude and behavior scores than those newly observed. The possible reason is that individuals diagnosed previously have already been educated by their doctors, but their behaviors have not yet changed, as changing dietary habits is a long-term process [38,39]. These results provide evidence for carrying out large-scale health education. In particular, efforts should be focused on reaching those people who have difficulty acquiring resources (i.e., rural individuals, the elderly, and less-educated people).

In China, there are more than 270 million individuals with hypertension and more than 330 million individuals with cardiovascular diseases [40]. Due to societal and economic development, the acceleration of urbanization, and the rapid transformation of lifestyles, these figures continue to increase at an alarming rate [40]. According to the latest dietary survey, the salt added by individuals during cooking was 9.3 g/d, accounting for 70–75% of the total salt intake [12,41]. It was estimated that reducing daily salt intake in China just 1 g could prevent almost 9 million cardiovascular events and save 4 million lives by 2030 [42]. Fully aware of the harm that high-salt diets cause and the benefits of salt reduction, the Chinese government has issued a series of policies [8]. In China, we need to integrate the resources available to the government and society so as to put forward more systematic, explicit, and practical salt reduction strategies and measures. Firstly, it is of paramount importance to carry out public health education, which is the foundation of behavior change. Strengthening evidence-based research on salt and health among Chinese people will increase public awareness of the dangers of excess salt consumption. At the same time, we should establish salt-reduction-support environments in schools, hospitals, restaurants, and other places. Secondly, home cooking is still the primary source of salt intake in China; thus, it is important to carry out salt reduction education through communities, grass-roots doctors, and other health professionals, as well as students. In northern China, for instance, a school-based salt reduction program successfully reduced the salt intake of children and their families by 25% [20]. In recent years, the food delivery service has been undergoing explosive growth in China [36]. More than one-fifth of the total population in China has already become a user of food delivery platforms [37]. Some studies have shown that males, young people, and the well-educated are increasingly dependent on food delivery [30,36,37,43]. Thus, educating and training restaurant chefs to reduce the amount of salt used in their dishes would aid in achieving the salt reduction goal. Specific measures to reduce salt in restaurants include posting “Less salt, More healthy” signs in all restaurants and training chefs to offer more low-salt options. Currently, the rapid growth in the consumption of pre-packaged food has contributed to 13.5% of total salt intake in China [13]. Setting incrementally lower salt targets has been shown to be effective in reducing the salt content in foods and has resulted in a reduction in population salt intake, as demonstrated by the UK and several other countries [16]. Moreover, there is much more the government can do to improve the nutrition labeling of pre-packaged food in China, such as the traffic light labeling implemented in the UK and the Chilean style warning labels [44,45,46]. In addition, the public should be informed and educated on how to read nutrition labels and choose healthy foods. What is important is that we need to establish a rigid and transparent surveillance system for salt reduction.

Our study had three main strengths. First, our study demonstrated that the association between salt-related KAB and 24-h urinary sodium was novel, as no study has previously investigated this relationship in China. Second, our study, for the first time, has gathered data from over 5300 participants across six provinces in the eastern, central, and western regions of China. Third, we utilized the most precise method for measuring 24-h urinary sodium excretion, which yielded a reliable estimate of salt intake.

The limitations of this study were as follows. First, the samples consisted of a single 24-h urine sample, which did not reflect the daily variation in salt intake and excretion. Second, our study included a larger proportion of rural participants, which may have led to an underestimation of salt-related KAB levels. Third, the validity and reliability of KAB score were tested in a pilot study with a smaller number of participants rather than systematically tested. Despite this, the consistent findings of KAB score and individual elements of the KAB questionnaires clearly indicate that individuals with higher KAB are more likely to have lower salt intake as measured by 24-h urinary sodium excretion. 

## 5. Conclusions

Our study showed that the Chinese population scored low for salt-related knowledge, attitude, and behavior, which was strongly associated with the 24-h urinary sodium results. Our findings indicate an urgent need for salt awareness education to increase individuals’ knowledge about salt reduction. Especially, efforts should focus on reaching older, less-educated people and those living in rural areas. Improving individuals’ awareness of salt reduction and decreasing their salt intake is a gradual process that needs to be constantly promoted for a long time. It is worth noting that the integration of multi-sectoral resources, such as the government, restaurants, communities, and schools, is required to tackle the challenge of salt reduction. With a joint effort from the whole of society, the continuous promotion of comprehensive salt reduction strategies will reduce salt intake in China.

## Figures and Tables

**Figure 1 nutrients-14-04404-f001:**
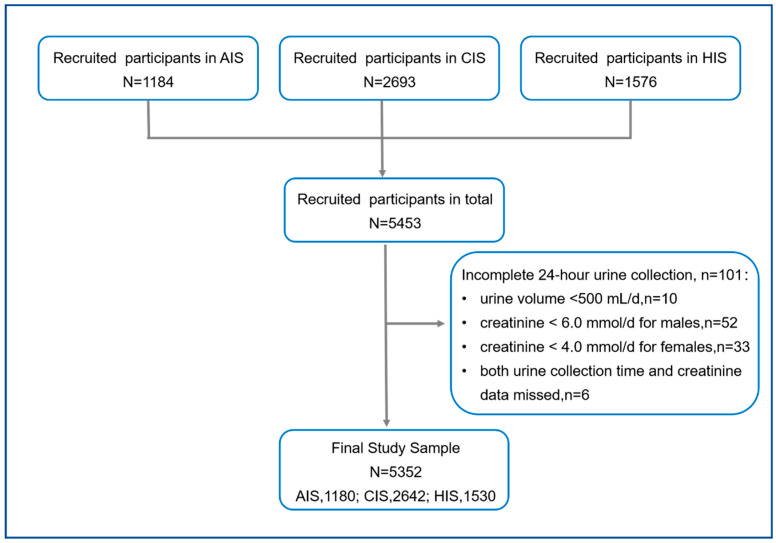
Flowchart of participants. We analyzed three independent RCTs for the Action on Salt China, including the AppSalt-based salt reduction program for primary school children and their families (AIS), the community-based comprehensive salt reduction intervention study (CIS), and the home cook salt reduction intervention study (HIS).

**Table 1 nutrients-14-04404-t001:** Characteristics of participants in the study (*n* = 5352).

Participants’ Characteristics	*n*	%
Location		
Urban	1437	26.8
Rural	3915	73.2
Sex		
Male	2601	48.6
Female	2751	51.4
Age, year; mean (SD)	49.76	12.84
Age groups; year		
18–44	2014	37.6
45–59	1863	34.8
≥60	1475	27.6
Education levels		
Low	2136	39.9
Medium	1800	33.6
High	1416	26.5
Hypertension *	1841	34.4
No	3511	65.6
Old diagnosed	1076	20.1
New observed	765	14.3
SBP, mm Hg; mean (SD)	125.62	19.16
DBP, mm Hg; mean (SD)	79.2	11.24
Scores; mean (SD)		
Score of knowledge	4.80	5.14
Score of attitudes	9.33	3.93
Score of behaviors	17.14	4.43
Score of overall KAB	31.27	9.18
Urinary volume, mL/24 h; mean (SD)	1602.97	650.98
Urinary creatinine, mmol/24 h; mean (SD)	10.62	3.23
Urinary sodium, mmol/24 h; mean (SD)	187.70	77.48

* Hypertension was defined as systolic blood pressure (SBP) ≥ 140 mmHg or diastolic blood pressure (DBP) ≥ 90 mmHg or taking anti-hypertensive drugs. “Old diagnosed” means previously diagnosed hypertension, and “new observed” means those observed at this screening.

**Table 2 nutrients-14-04404-t002:** Description of knowledge, attitude, and behavior related to salt.

Questions	*n*	%
Knowledge (3 questions)		
Do you know what is the recommended maximum level of salt per day?		
Less than 2 g	114	2.13
Less than 5 g *	1198	22.38
Less than 8 g	108	2.02
Less than 12 g	78	1.46
More than 12 g	22	0.41
Don’t know	3832	71.60
Have you heard of low-sodium salt?		
Yes *	1638	30.61
No	3714	69.39
Which item on this food label represents the salt content?		
Energy	103	1.93
Protein	60	1.12
Fat	29	0.54
Carbohydrates	225	4.20
Sodium *	2305	43.07
Don’t know	2630	49.14
Attitudes (3 questions)		
Do you agree high salt intake would cause hypertension?		
Agree *	4079	76.21
Disagree	406	7.59
Don’t know	867	16.20
Do you agree low salt intake would make people limb weakened?		
Agree	2965	55.40
Disagree *	1513	28.27
Don’t know	874	16.33
Would you like to choose a lower-salt diet?		
Yes *	4389	82.01
No	907	16.95
Don’t know	56	1.04
Behaviors (6 questions)		
What is your usual taste for food?		
More Salty	1587	29.65
Moderate	2327	43.48
Less Salty	1438	26.87
Do you use low-sodium salt at home?		
Yes	556	10.39
No	835	15.60
Don’t know	3961	74.01
How often have you consumed pickled foods in the past month?		
Almost every day	624	11.66
3–5 days per week	433	8.09
1–2 days per week	1046	19.54
Once per week or less	3249	60.71
How often have you consumed salty snacks in the past month?		
Almost every day	39	0.73
3–5 days per week	68	1.27
1–2 days per week	327	6.11
Once per week or less	4918	91.89
How often did you eat out or order delivery in the past month?		
Almost every day	198	3.70
3–5 days per week	348	6.50
1–2 days per week	705	13.17
Once per week or less	3013	56.30
Never	1088	20.33
Have you requested for less-salted meals when eating out in the past month? (n = 4264) ^a^		
Always	275	6.45
Sometimes	226	5.30
Occasionally	228	5.35
Never	3535	82.90

* Regarded as the right/positive answer for salt-related knowledge/attitude questions. ^a^ Those who have never eaten out or ordered delivery in the past month were not included.

**Table 3 nutrients-14-04404-t003:** Knowledge, attitude, behavior, and KAB scores of participants with different sociodemographic characteristics.

Variables	Score of Knowledge			Score of Attitudes			Score of Behaviors			Score of KAB		
	Mean ± SD	t/F	*p*-Value	Mean ± SD	t/F	*p*-Value	Mean ± SD	t/F	*p*-Value	Mean ± SD	t/F	*p*-Value
Location												
Urban	7.25 ± 5.21	21.062	<0.001	10.36 ± 3.79	12.056	<0.001	17.98 ± 4.67	8.313	<0.001	35.59 ± 8.83	21.963	<0.001
Rural	3.93 ± 4.82			8.94 ± 3.92			16.80 ± 4.31			29.64 ± 8.81		
Sex												
Male	4.99 ± 5.08	2.241	0.025	8.79 ± 4.14	−9.726	<0.001	16.54 ± 4.49	−9.319	<0.001	30.27 ± 9.35	−7.458	<0.001
Female	4.67 ± 5.21			9.83 ± 3.66			17.67 ± 4.32			32.13 ± 8.96		
Age groups												
18–44	6.56 ± 5.16	222.543	<0.001	9.88 ± 3.97	32.200	<0.001	16.27 ± 4.38	97.452	<0.001	32.65 ± 9.49	39.706	<0.001
45–59	4.32 ± 4.96			9.02 ± 3.88			17.07 ± 4.40			30.36 ± 9.17		
≥60	3.09 ± 4.59			8.95 ± 3.87			18.36 ± 4.30			30.37 ± 8.56		
Education levels												
Low	1.74 ± 3.19	1353.959	<0.001	8.57 ± 3.75	112.157	<0.001	17.39 ± 4.22	18.544	<0.001	27.69 ± 7.18	547.889	<0.001
Medium	4.99 ± 4.67			9.24 ± 4.05			16.60 ± 4.37			30.77 ± 8.95		
High	9.22 ± 4.81			10.55 ± 3.74			17.38 ± 4.77			37.11 ± 9.29		
Hypertension												
No	5.34 ± 5.22	57.901	<0.001	9.43 ± 4.03	5.064	0.006	17.05 ± 4.43	4.838	0.008	31.77 ± 9.38	18.023	<0.001
Old diagnosed	4.26 ± 4.98			8.94 ± 3.93			16.97 ± 4.50			30.16 ± 9.20		
New observed	3.55 ± 4.73			9.25 ± 3.58			17.51 ± 4.42			30.32 ± 8.40		

**Table 4 nutrients-14-04404-t004:** Univariate and multivariate analysis (regression coefficient and *p*-value) of salt-related KAB scores.

Variables	Univariate Analysis		Multivariate Analysis	
	β (95% CI)	*p*-Value	β (95% CI)	*p*-Value
Location				
Urban	ref		ref	
Rural	−5.924(−6.458, −5.390)	<0.001	−1.165(−2.488, −0.150)	0.031
Sex				
Male	ref		ref	
Female	1.857(1.364, 2.350)	<0.001	2.734(2.291, 3.186)	<0.001
Age groups				
18–44	ref		ref	
45–59	−2.297(−2.870, −1.719)	<0.001	0.437(−0.129, 0.997)	0.129
≥60	−2.296(−2.913, −1.680)	<0.001	1.350(0.696, 2.002)	<0.001
Education levels				
Low	ref		ref	
Medium	3.140(2.612, 3.669)	<0.001	3.802(3.255, 4.342)	<0.001
High	9.453(8.888, 10.018)	<0.001	9.266(8.628, 9.902)	<0.001
Hypertension				
No	ref		ref	
Old diagnosed	−1.638(−2.350, −0.923)	<0.001	−0.345(−0.997, 0.312)	0.302
New observed	−1.484(−2.110, −0.859)	<0.001	−0.037(−0.637, 0.567)	0.905

**Table 5 nutrients-14-04404-t005:** Association of salt-related knowledge and attitude scores and the behavior score.

Variables	Univariate Analysis		Multivariate Analysis *	
	β (95% CI)	*p*-Value	β (95% CI)	*p*-Value
Score of knowledge	0.102(0.080, 0.126)	<0.001	0.143(0.116, 0.169)	<0.001
Score of attitudes	0.214(0.184, 0.243)	<0.001	0.178(0.149, 0.207)	<0.001

* Multivariate analysis adjusted for location (rural/urban), sex, age groups, education levels, and hypertension (no/old diagnosed/new observed).

**Table 6 nutrients-14-04404-t006:** Association between salt-related KAB scores and 24-h urinary sodium excretion.

Variables	Models	β (95% CI)	*p*-Value
Score of knowledge	model 1	−1.046 (−1.478, −0.634)	<0.001
	model 2	−1.456 (−1.893, −1.038)	<0.001
	model 3	−0.800 (−1.285, −0.312)	0.001
Score of attitudes	model 1	−0.850 (−1.388, −0.323)	0.002
	model 2	−0.595 (−1.132, −0.067)	0.029
	model 3	−0.284 (−0.816, 0.247)	0.295
Score of behaviors	model 1	−2.697 (−3.175, −2.220)	<0.001
	model 2	−2.275 (−2.760, −1.792)	<0.001
	model 3	−2.200 (−2.673, −1.716)	<0.001
Scores of overall KAB	model 1	−1.124 (−1.364, −0.897)	<0.001
	model 2	−1.069 (−1.306, −0.841)	<0.001
	model 3	−0.851 (−1.095, −0.602)	<0.001

Model 1 was an unadjusted result. Model 2 adjusted for sex and age groups. Model 3 adjusted for location (rural/urban), sex, age groups, education levels, and hypertension (no/old diagnosed/new observed).

## Data Availability

Not applicable.

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
