# Peer review of "Salt-Related Knowledge, Attitudes, and Behaviors and Their Relationship with 24-Hour Urinary Sodium Excretion in Chinese Adults"

_nutrients, 2022, doi:10.3390/nu14204404_

Round 1

Reviewer 1 Report

Thank you for the opportunity to review this interesting paper that assesses dietary sodium KAB in a population of Chinese adults who were enrolled in large population based clinical trials, and to examine the relationship between sodium KAB and salt intake in these individuals.

Overall, this paper is very well written and presented. The findings are interesting, novel and highly relevant to dietary sodium policy development.

Background

The background provides appropriate context for the paper. The background could be strengthened by highlighting the research gap in this area of study to strengthen the justification and potential impact for this study. The objectives are clear.

Methods

The methods are well written. Some increased clarity is required, mostly in Section 2.2 – Survey Instrument.  

Line 86 – Knowledge and attitudes were scored 5 points for each “correct” answer.  It is unclear how an attitude be correct/incorrect. Please clarify.

Line 88 and 89 – Knowledge and Attitude scores were scored out of 15, and behaviour scores out of 30. The overall weighting of the scores is based on the number of questions, however, this isn’t immediately obvious. Suggest the authors revise these sentences to make the rationale for the weights more salient.

Line 89 and 90 – “best positive response” and “worst negative response” are not good descriptors of sodium-related behaviours. Suggest the use of more clear terms.  

Line 100 and 101 – If the 24-hour urine collections were <500 mL, or if creatinine excretion was <4.0 mmol for women or <6.0 mmol for men, were the urine collections excluded? One assumes so, but this must be stated. Was “over collection” of the urine collection assessed? Finally, the reference standards refer to men/women (gender-based); however sex (based on body size) is a more likely determinant of creatine excretion. Please correct the use of sex and gender-based language.

Line 107 – Please change “primary school education or illiteracy” to “primary school education or less”.

Results

Line 123 - How many of the urine collections (if any) were excluded because they did not meet the criteria for completeness?

Table 1/2 and throughout the manuscript – Please be clear on if you have referring to sex or gender, and ensure this is consistent throughout the manuscript (including your interpretation of papers in the Discussion). Sex/Gender consideration. In Table 1 variable name is Gender, but then male/female (sex) is listed. However, in Table 2 the variable name is sex, and male/female (sex) are used. Sodium-related KAB is more likely to be influence by gender (i.e., socially constructed roles, expressions, behaviours) and thus gender as a co-variate is be appropriate. However, gender (man/woman/gender minorities) must have been ascertained by participants themselves during the data collection phase. If gender was not used, please discuss the use of sex as a potential study limitation.

Section 3.2 – Interesting, sodium KAB in this population is quite low. Only the “composite” scores for sodium-related KAB have been reported. The individual questions/responses for sodium KAB have not been included. This would help the reader have a deeper understanding of how the individual scores were derived for each of the 12 items on sodium-related KAB.

Discussion

Line 262 – In this study you linked an individual’s sodium excretion with their KAB. The argument that pervious research could be used to estimate group sodium intake levels is irrelevant, as that was not the objective of this study.

Please include the limitation that the individuals in this study were enrolled in randomized trials, and thus the findings may not be generalizable to the general population.  

Editorial

Please use sodium “KAB” acronym consistently throughout the paper.
Line 151 – fix the grammatical error/typo.

Reviewer 2 Report

The manuscript "Salt-related knowledge, attitudes, and behaviors and their relationship with 24-hour urinary sodium excretion in Chinese adults" addresses an important public health issue, is concisely and clearly written, and analyzes a numerically large sample in which 24-hour urine natriuresis was assessed to estimate salt intake. However, I have some comments to make to the authors.

- In my opinion, the questionnaire should be reported in full in the main manuscript and not in the supplementary material, because it allows the reader to better understand what aspects were investigated, especially what "attitude" means, which in the current version of the manuscript is not very clear. I would also like to ask the authors if and how the questionnaire and related score were validated. I think this is a key point, as the type of questions and the score assigned to each are the basis for the whole statistical analysis of the paper. In particular, for the knowledge and attitude items, 5 points are awarded for each correct answer, so even one more correct answer changes the score significantly. In the behavior part, a score is acquired even for only partially correct answers. I wonder if this different weight given to the divesrse questions of the questionnaire, might not create an imbalance between its three parts.

- By what formula was sodium intake estimated from the 24-hour urine sodium content? A single 24-hour sodium assay, as the authors also point out in the limitations, is indicative only of the previous day's salt intake and not of the habit of eating more or less salt. However, the size of the sample may obviate this possible inaccuracy. I think, however, that it would have been more appropriate to collect urine before administering the questionnaire, because it is clear from the questions asked that the study wanted to investigate salt consumption by giving it a negative connotation, and because of this, some of the respondents may have limited their salt intake on the day of urine collection.

- The authors do not comment on the data reported in Table 2 regarding the presence or absence of hypertension. Hypertensive individuals have less knowledge of the negative effects of salt intake and a worse overall score than normotensive individuals. How many of the patients defined as hypertensive were diagnosed at the time of screening ( and then it would be reasonable for them to have little knowledge of the problems related to excess salt intake) and how many were hypertensive already diagnosed as such? The latter group should have already been educated by their physician regarding the problems arising from excessive salt intake. Please divide the hypertensives into two groups, new and old diagnosis, and analyze them as two separate variables.

- Another interesting finding in Table 2 is the gender difference in the responses to the questionnaire. Males would have, albeit slightly, better knowledge of the problem, but worse attitudes and behaviors. This might make the discussion statement, "individuals' dietary preferences and behaviors are influenced by their knowledge and attitudes" (from lines 196-7) questionable. It appears from Table 4 that behavior greatly influences the amount of urinary sodium and thus salt intake, while the effects of knowledge and attitudes are significantly more modest. To show how much knowledge and attitudes influence behavior, I think a statistical analysis that establishes the relationship between the score of the first six questions of the questionnaire (knowledge and attitude) with the second six (behavior) would be useful.

- From Table 3 it would appear that at the uni-factor analysis, age has a positive association on the score, while this is reversed in the multi-factor analysis. From Tables S2, S3, S4 it seems that, if I interpret the data correctly, younger people have better knowledge and attitudes but worse behavior than older people. This fact should be emphasized and discussed in the discussion.

Round 2

Reviewer 1 Report

Thank you for an excellent revision. There is one suggestion that was missed:

"Nonetheless, as demonstrated by previous research, one 24-hour urine collection could still be used to estimate group levels." This sentence is irrelevant to the current study and should be removed, as the objective was to assess individual intakes, not group intakes. 

Author Response

1. There is one suggestion that was missed: "Nonetheless, as demonstrated by previous research, one 24-hour urine collection could still be used to estimate group levels." This sentence is irrelevant to the current study and should be removed, as the objective was to assess individual intakes, not group intakes.

Our response: Thank you for taking time to review our paper again. We agree with your opinion, so we have deleted it now.

Reviewer 2 Report

The authors' answers and the changes made to the manuscript regarding the point of validation of the questionnaire and the scores given to the different questions do not seem satisfactory to me. In fact, I feel that a 'round of expert consultation to discuss whether it is reasonable' is not sufficient and the references suggested in the reply for the reviewers do not seem entirely relevant and do not help to resolve my concerns. In the absence of a validated test, at least a pilot study would have been necessary. Since the scores obtained from the answers to the questionnaire form the basis of the statistical analysis and the conclusions drawn from it, it is reasonable to assume that even small differences in the questions or in the scores assigned could have led to different results.This is, in my opinion, an important limitation of the study.

In the reply to the reviewers, the formula for calculating the assumed salt is incomplete, whereas in the revised manuscript it is correct. However, the bibliographic source (ref 28) of the formula refers to a study in which the formula was used and not to the original study in which it was proposed and validated. In the paper cited by ref 28, the authors refer to a meta-analysis that considers three formulas namely INTERSALT, TANAKA, and KAWASAKI (International Journal of Epidemiology, 2016, 239-250 doi: 10.1093/ije/dyv313), but again this is not a study that validated a specific formula. I would like to understand which of the three formulas (INTERSALT, TANAKA, or KAWASAKI ) was used by the authors and what was the reason for the choice.  

A recent article in Nutrients (Nutrients 2022, 14, 2736. https://doi.org/10.3390/nu14132736) compared four formulas estimating sodium intake (KAWASAKI, INTERSALT, TANAKA, and SUN). The authors conclude that the two most valid formulas are the INTERSALT and the KAWASAKi and that INTERSALT method provides the best estimation of the population's mean sodium intake in summer and the Kawasaki method is superior to other methods in winter. If the authors used one of these two formulae, a comment should be made in consideration of the season of the year in which the authors collected their data.

I suggest that the table on the relationship between knowledge and behaviour should be put in the main manuscript and not as a supplementary.

The answers to the other questions raised by this reviewer and the related changes to the manuscript are satisfactory. 

Author Response

Thank you for taking time to review our paper again. Please see the attachment.
